Hanging under the ledge: synergistic consequences of UVA and UVB radiation on scyphozoan polyp reproduction and health

Johnson Lauren E. 1 2
Treible Laura M. laura.treible@gmail.com 1 3
1 Institute for Coastal Plain Science, Georgia Southern University , Statesboro , GA , United States of America
2 Department of Biology, Washington University in St. Louis , St. Louis , MO , United States of America
3 Skidaway Institute of Oceanography, University of Georgia , Savannah , GA , United States of America
Pogoreutz Claudia
Electronic publication date: 2023 Feb 2
Publication date: 2023
Volume: 11
Electronic Location ID: e14749
Received 2022 Aug 19; Accepted 2022 Dec 27
Copyright: ©2023 Johnson and Treible
Copyright year: 2023
Copyright holder: Johnson and Treible
License: This is an open access article distributed under the terms of the Creative Commons Attribution License, which permits unrestricted use, distribution, reproduction and adaptation in any medium and for any purpose provided that it is properly attributed. For attribution, the original author(s), title, publication source (PeerJ) and either DOI or URL of the article must be cited.
License URL: https://creativecommons.org/licenses/by/4.0/

Keywords: Scyphozoan, Ultraviolet radiation, Cnidarian, Polyp, Asexual reproduction

Funding: The National Science Foundation (NSF) Research Experience for Undergraduates (REU) at Georgia Southern University’s Institute for Coastal Plain Science 1757536 This work was supported by the National Science Foundation (NSF) Research Experience for Undergraduates (REU) at Georgia Southern University’s Institute for Coastal Plain Science (Award No. 1757536). The funders had no role in study design, data collection and analysis, decision to publish, or preparation of the manuscript.

==============================
Overexposure to ultraviolet radiation (UVR) emitted by the sun can damage and kill living cells in animals, plants, and microorganisms. In aquatic environments, UVR can penetrate nearly 47 m into the water column, severely impacting many marine organisms. Jellyfish are often considered resilient to environmental stressors, potentially explaining their success in environmentally disturbed areas, but the extent of their resilience to UVR is not well known. Here, we tested resiliency to UVR by exposing benthic polyps of the moon jellyfish, Aurelia sp., to UVA and UVB—the two types of UVR that reach Earth’s surface—both separately and in combination. We quantified asexual reproduction rates and polyp attachment to hard substrate, in addition to qualitative observations of polyp health. There were no differences in asexual reproduction rates between polyps exposed to isolated UVA and polyps that received no UVR. Polyps reproduced when exposed to short term (∼7–9 days) isolated UVB, but long-term exposure limited reproduction and polyp attachment to the substrate. When exposed to both UVA and UVB, polyps were unable to feed and unable to remain attached to the substrate, did not reproduce, and ultimately, experienced 100% mortality within 20 days. Although many studies only examine the effects of UVB, the combination of UVA and UVB here resulted in greater negative impacts than either form of UVR in isolation. Therefore, studies that only examine effects of UVB potentially underestimate environmentally relevant effects of UVR. These results suggest that polyps are unsuccessful under UVR stress, so the planula larval stage must settle in low-UVR environments to establish the success of the polyp stage.

Introduction

Ultraviolet radiation (UVR) emitted by the sun, which contains both UVA (320–400 nm) and UVB (290–320 nm) radiation, penetrates deep within living tissues, and overexposure can damage and kill cells (Setlow, Swenson & Carrier, 1963; Karentz, Cleaver & Mithcell, 1991; Buma et al., 1995). Most organisms must balance reducing overexposure to UVR while continuing to forage, mate, avoid predators, and other ecological activities. Although oceans cover two-thirds of Earth’s surface, UVR was initially assumed to have minimal impact on marine life because UVR can rapidly attenuate in natural waters (Smith & Baker, 1979). However, these initial data underestimated the penetration depth of UVR (Morel et al., 2007). UVR can penetrate up to 22 m in clear coastal waters, and almost 50 m in open ocean (Tedetti & Sempéré, 2006). Notably, DNA-damaging doses of UVR can reach depths up to 4.5 m and 12 m in clear coastal waters and open ocean, respectively (Tedetti & Sempéré, 2006).

Increased exposure to UVB radiation leads to increased mortality in a variety of marine organisms (Llabrés et al., 2013). However, sensitivity to UVR varies by taxa, with cnidarians, including scyphozoan jellyfish, showing lesser sensitivity to changes in UVB than other organisms (Llabrés et al., 2013). Across systems, scyphozoan jellyfish have shown resilience to a variety of environmental stressors (Richardson et al., 2009), including warming ocean temperatures (Purcell, 2007; Purcell, Hoover & Schwarck, 2009), coastal acidification (Winans & Purcell, 2010; Enrique-Navarro et al., 2021), increased occurrences of hypoxia (Condon, Decker & Purcell, 2001; Grove & Breitburg, 2005; Thuesen et al., 2005; Miller & Graham, 2012), as well as combinations of acidification and hypoxia (Treible et al., 2018). Their resilience has been used as one explanation for the apparent increase in jellyfish blooms in many coastal waters around the globe (Richardson et al., 2009). However, the impact of UVR on the ecology and success of jellyfish populations is not well understood.

What little we know about UVR’s effect on jellyfish ecology has been primarily derived from studies on freshwater medusae (Salonen et al., 2012; Caputo et al., 2018). Medusae are the free-swimming sexually reproducing stage of cnidarians. Overexposure to solar radiation has been shown to impair swimming responses and reduce survival (Caputo et al., 2018), and some species may die within just one hour of unshielded exposure (Salonen et al., 2012).

Understanding how medusae respond to UVR can help explain distribution patterns in adult jellyfish. However, many jellyfish also have an often-overlooked early life stage of benthic polyps. Polyps are formed after sexually produced planula larvae settle onto benthic substrates where they eventually metamorphose (Lucas, 2001; Lucas, Graham & Widmer, 2012). Polyps then asexually reproduce in two ways: they can clone themselves through budding new polyps (Schiariti et al., 2014), and they can strobilate, a process that produces pelagic ephyrae (juvenile jellyfish) that grow into the adult medusae. Polyps have the ability to strobilate multiple times per year (Purcell, 2007), and certain species can produce over thirty ephyrae per polyp in a single strobilation event, depending on environmental conditions such as temperature (reviewed in Purcell et al., 2012). Therefore, a single polyp, through cloning and strobilation, can have exponential impacts on the resulting adult population (Lucas, 2001; Lucas, Graham & Widmer, 2012).

Marine organisms have evolved a range of strategies to mitigate UVR-induced damage, which may vary depending on life history. Negative effects of UVR would be most evident for pelagic stages that are directly exposed to radiation in surface waters, but pelagic organisms could avoid UVR by migrating vertically to deeper areas in the water column (Salonen et al., 2012). Another potential strategy is the presence of UV-absorbing pigments including mycosporine-like amino acids (MAAs), that can act as a “sunscreen” for UVR (Nakamura, Kobayashi & Hirata, 1982; Dunlap & Chalker, 1986). MAAs can be found in a suite of invertebrates and can either be acquired through diet (Banaszak & Trench, 1995; Carroll & Shick, 1996; Carefoot et al., 1998; Helbling, Menchi & Villafañe, 2002; Häder et al., 2007) or through relationships with MAA-containing symbionts (Banaszak & Trench, 1995; Dunlap & Shick, 1998; Torres-Pérez & Armstrong, 2012). Some scyphozoan jellyfish, such as the upside-down jellyfish Cassiopea sp., have zooxanthellae symbionts that require exposure to light for photosynthesis, but may also provide MAAs for UVR protection (Klein, Pitt & Carroll, 2016). Additionally, some studies suggest that the development of UVR coping strategies may originate from exposure to UVR in benthic early life stages (Fischer & Phillips, 2014), and likewise, organisms routinely shielded from UVR may be more susceptible to negative impacts.

Field and lab studies consistently demonstrate that 80–100% of planulae preferentially settle on the underside of substrates, resulting in polyps that are attached upside-down (Svane & Dolmer, 1995; Pitt, 2000; Holst & Jarms, 2007). There are several possible explanations for this behavior. Attaching upside-down allows gravity to support defecation and could thus prevent mortality from decaying food residues in the gastric cavity (Holst & Jarms, 2007). Residing upside-down could also make the production and release of ephyrae by strobilation more efficient (Brewer, 1976). This behavior may also be an evolutionary adaptation to avoid mortality by sedimentation, predation, or solar irradiation (Svane & Dolmer, 1995). The proliferation of engineered artificial structures in shallow coastal environments (“ocean sprawl”) has created novel habitats for polyps (Duarte et al., 2013) that can impact resulting medusae populations (Makabe et al., 2014; Feng et al., 2017), yet distributions of polyps on these structures may be influenced by the penetration of solar irradiation inherent in relatively shallow coastal waters.

The objective of this study was to determine the impacts of solar irradiation, particularly UVR, on jellyfish polyps, as one potential mechanism driving planulae/polyps to preferentially settle on the underside of substrates. We examined the response of polyps of the moon jellyfish, Aurelia sp., a widespread species whose populations occur along most global coastlines (Lucas, 2001; Collins, Jarms & Morandini, 2018) and are known to tolerate a wide range of environmental factors (Schiariti et al., 2014). We determined the effects of UVA, UVB, and combined UVA and UVB radiation on polyp survival, asexual reproduction, and health.

Based on previous UVR studies in freshwater medusae (e.g., Salonen et al., 2012; Caputo et al., 2018) and enhanced mortality rates of marine organisms associated with elevated UVB (Llabrés et al., 2013), we hypothesized that polyp survival and reproduction would be hindered by exposure to UVB. We also hypothesized that the addition of UVA in combination with UVB—which is more environmentally relevant than just UVB alone—would exacerbate observed negative effects on polyps. The only previous study on the response of scyphozoan polyps to UVR only included UVB wavelengths and suggested an inconsistent response of polyps to changes in UVB (Klein, Pitt & Carroll, 2016). Yet UVA can penetrate deeper into the water column (Tedetti & Sempéré, 2006), and should, therefore, not be ignored (Kligman, Akin & Kligman, 1985).

Methods

Experiment set up

Aurelia sp. polyps were shipped from the Shedd Aquarium (Chicago, IL, USA) to Georgia Southern University (Statesboro, GA, USA) in June 2019. Upon arrival, the polyps remained in their original seawater (salinity 30) and were allowed to acclimate to experimental temperature of 20 °C. After 3 h, we transferred the polyps and their original seawater into a clean glass beaker, added approximately 50% Instant Ocean artificial seawater (salinity 35), and fed the polyps freshly hatched brine shrimp, Artemia spp. Polyps settled and attached to the bottom and walls of the glass beaker, establishing a culture population for the subsequent experiments.

After 24 h, we haphazardly selected polyps from the culture, gently removed them with a 1 mL plastic pipette, and transferred them individually into eighty-nine 40 mm diameter plastic culture dishes (n = 1 polyp per dish) that contained Instant Ocean artificial seawater (salinity 35). Polyps acclimated to the experimental conditions (35 and 20 °C) for 5 days prior to the experiment, during which time they settled and attached to the dishes. We monitored the polyps daily to ensure attachment and general overall health.

We did not include dishes with unattached polyps in the experiment, and we removed extra polyps at the start of the experiment in any dishes with >1 polyp (a result of asexual budding). We randomly assigned experimental dishes (n = 54) containing single attached polyps to eighteen 0.5 L glass tanks (n = 3 dishes per tank). The tanks contained Instant Ocean artificial seawater (35) and were kept gently aerated with air stones. We occasionally topped up the tanks with artificial seawater (35) to replace volume lost to evaporation, and when necessary, exchanged 25% to 75% of the seawater in each tank to maintain consistent salinity (35.3 ± 1.2). Air temperature in the laboratory was maintained at a constant temperature, which resulted in consistent experimental water temperature (20.4 ± 0.5 °C) throughout the experiment.

We constructed three independent light chambers separated by black plastic sheeting to test UVR exposure on polyp reproduction and health (Fig. 1). Each chamber contained two fluorescent bulbs: UVA only consisted of two Philips F20T12/BL 20-Watt bulbs; UVB only consisted of two Light Sources FS20T12/UVB 20-Watt bulbs; and UVA + UVB consisted of one Philips F20T12/BL 20-Watt bulb and one Light Sources FS20T12/UVB 20-Watt bulb.

Figure 1 Experimental setup.

Three independent light chambers were separated by black plastic sheeting; Left: UVB only (dark gray), two Light Sources FS20T12/UVB 20-Watt bulbs; Middle: UVA only (light gray), two Philips F20T12/BL 20-Watt bulbs; Right: UVA + UVB (medium gray), one Philips F20T12/BL 20-Watt bulb and one Light Sources FS20T12/UVB 20-Watt bulb. Six 1.9 L glass tanks were placed within each light chamber; four tanks were uncovered and exposed to the experimental UVR, while two tanks in each light chamber were covered with a Lexan polycarbonate sheet to filter and remove all wavelengths of UVR and to represent ‘no UVR’ controls (lightest gray). Each jar contained three dishes (n = 1 polyp per dish). Polyps were exposed to UVR to a 6:18 h light:dark cycle. Light intensities were based on ambient levels measured in Statesboro, Georgia (Table 1).

Each light chamber held six 1.9 L glass tanks. Four tanks were uncovered and exposed to the experimental UVR, while two tanks in each light chamber were covered with a Lexan polycarbonate sheet to filter and remove all wavelengths of UVR and to represent ‘no UVR’ controls. Polycarbonate excludes all wavelengths below 400 nm, blocking both UVB and UVA transmission (e.g., Gregan et al., 2012).

We based experimental UVR levels on ambient levels measured during the summer in Statesboro, Georgia (32.44°N, −81.77°W). We recorded average instantaneous irradiance (µW cm−2) at 379.68 nm (UVA) and 305.22 nm (UVB) between midday and peak sun hours (10:00 to 16:00 local time) on June 13, 2019 with an OCR-504 UV Multispectral Radiometer (Satlantic). We performed all recordings in about 14.5 cm of artificial seawater to imitate the potential depth of novel substrates found in shallow coastal waters and to reflect the intensity of UVR that polyps would experience in the experimental tanks.

We calculated the average instantaneous irradiation levels between 10:00 and 14:00 local time (Table 1) and manually adjusted the distance between the light bulbs and the tanks to achieve instantaneous UVA and UVB irradiation levels that best reflected peak UVR exposure levels.

Table 1 Instantaneous and total intensity (µW cm−2) of ambient and experimental UV conditions measured at 379.68 nm (UVA) and 305.22 nm (UVB).

Mean ± S.D. Ambient intensities shown are the average instantaneous intensity between 10:00 to 14:00 and the total intensity experienced between 10:00 to 16:00 on June 13, 2019 in Statesboro, Georgia. Treatment intensities are the average of three separate recordings of instantaneous intensity under each treatment, and the total intensity calculated from the respective instantaneous intensity over a sum of 6 hours of treatment exposure.

	Average instantaneous intensity (µW cm−2)	Total intensity (µW cm−2)	
Treatment	379.68 nm (UVA)	305.22 nm (UVB)	379.68 nm (UVA)	305.22 nm (UVB)	
Ambient 	30.8 ± 4.98	1.45 ± 0.47	176.86	8.42	
UVA only 	1.89 ± 0.02	−0.05 ± 0.00	11.34 ± 0.10	−0.30 ± 0.01	
UVB only 	0.02 ± 0.00	1.45 ± 0.01	0.14 ± 0.00	8.71 ± 0.07	
UVA + UVB	1.94 ± 0.08	1.44 ± 0.02	11.65 ± 0.49	8.65 ± 0.15	
Controls 	0	0	0	0	

While we aimed for environmentally relevant daily doses of UVR, our treatments containing UVA radiation experienced lower than ambient radiation in the UVA range. We achieved ambient UVB levels at 305.22 nm in both the UVB only and UVA + UVB treatment (Table 1). However, when we adjusted the height of the light fixture in the combined UVA + UVB treatment to achieve ambient UVB levels at 305.22 nm, the irradiance at 379.68 nm (UVA) was below ambient levels. If we lowered the height of the fixture (i.e., increasing the UVA intensity), we would have subsequently also increased UVB intensity above ambient levels. We, therefore, adjusted the intensity of the UVA only treatment to match the irradiance at 379.68 nm in the combined UVA + UVB treatment so that the levels of UVA experienced would be comparable in both treatments (Table 1).

We exposed polyps to the UVR light treatments on a 6:18 h light:dark cycle to mimic the length of midday peak sun hours (10:00 to 16:00; 6 h). The light cycle occurred during 23:00 to 05:00 local time to ensure that measurements and feeding would not occur during the UVR light cycle and to minimize personnel exposure to UVR.

Data collection

We collected data on polyp asexual reproduction and health daily for the first week, then every other day for a total of 27 days. On each data collection day, we gently removed polyp dishes from each experimental tank. We quantified asexual reproduction (i.e., budding) by counting the total number of polyps in each dish. From this count data, we calculated a weekly budding rate: Weekly Budding Rate=Nfinal−NinitialTotal Days×7days

where Ninitial is the number of polyps at the beginning of the experiment (n = 1), Nfinal is the number of polyps at the end of the experiment (number of polyps on day 27), and Total Days is the duration of the experiment (27 days). Any calculated negative budding rates due to excess mortality were considered zero.

After completing asexual reproduction data collection, we fed the polyps freshly hatched brine shrimp, Artemia spp., and allowed them to feed for 1–2 h. During feeding, we observed the polyps under a Leica dissecting microscope and recorded detailed qualitative notes on the general appearance of the polyps (e.g., if they appeared “healthy” or “unhealthy”). We defined a “healthy” polyp as a polyp with extended tentacles that actively fed and remained attached to the dish. We recorded if any polyps appeared “unhealthy”, including if a polyp had retracted or absent tentacles, had no presence of food in the gut (evidence of not feeding), and/or did not remain attached within the dish. For quantitative purposes, we only used detachment as a proxy for health, where polyps that became detached from their substrate were deemed unhealthy. To account for potential accidental detachments due to handling, we only considered polyps that were detached for more than one time point; if a polyp became detached but then subsequently reattached to the dish, we did not consider it detached in the analysis. Polyp detachment was classified binomially as either yes (polyps in dish detached at >1 timepoints) or no (polyps in dish detached at ≤ 1 timepoints).

While polyps were feeding, we also measured temperature and salinity in each tank and replaced or exchanged water, when necessary, as previously described. After feeding, we removed excess brine shrimp and other feeding debris with a 1 mL plastic pipette before gently returning the dishes to their treatment tanks. We then randomly shuffled tanks within each treatment group under their respective light fixtures as they were replaced, to minimize any spatial effects of UVR variability (i.e., potential for lesser intensity near the ends of the bulbs). We performed all data collection and feeding in low natural light during the dark cycle of the UVR exposure.

Data analysis

All statistical analyses were performed using R (version 4.1.1) in RStudio (Version 1.2.1335; RStudio Team, 2020). To compare the independent and interactive effects of UVA and UVB on weekly budding rate, we first ran a two-way ANOVA with UVA and UVB as the main effects. Normality of residuals was confirmed using quantile–quantile plots (Wilk & Gnanadesikan, 1968), and although weekly budding rate did not meet the parametric assumption of homogeneity of variances, this was to be expected as the combined UVA + UVB treatment experienced 100% mortality. A post-hoc Tukey test was used when significant differences were detected to determine pairwise comparisons between treatments. We also ran a chi-squared test of independence to test whether there were non-random differences in detachment probabilities across the treatment groups.

Results

Polyp reproduction under UVR

Weekly budding rates differed among the UVR treatments (Fig. 2B, Tables S1, S2). Polyps exposed only to UVA radiation had the highest average budding rate (1.60 ± 0.85 buds week−1), but it did not differ significantly from the budding rate of control polyps not exposed to UVR (1.14 ± 0.92 buds week−1; Fig. 2B, Tables S1, S2). The UVA only polyps and the control polyps also both ended the experiment on day 27 with similar numbers of polyps (UVA: 6.62 ± 1.07 polyps; control: 5.39 ± 0.82 polyps; Fig. 2A). Polyps exposed only to UVB successfully budded but more slowly, at less than half of the rate of UVA only polyps (0.65 ± 1.10 buds week−1; Fig. 2B, Tables S1, S2). The UVB only treatment ended with an average of 3.5 ± 1.22 total polyps on day 27 (Fig. 2A). Polyps exposed to both UVA and UVB did not bud at all during the study and had 100% polyp mortality by day 21 (Fig. 2).

Figure 2 Polyp reproductive response to UVR exposure.

(A) Total number of polyps observed (mean ± s. e.) under each UV treatment over the 27-day experiment. (B) Weekly budding rates (polyps/week) for each UV treatment. ‘X’ indicates the mean. Lowercase letters (a, b, c) indicate significant differences (p < 0.05) between groups based on ANOVA with Tukey’s post hoc tests. Treatment details are given in Table 1.

Polyp health and detachment under UVR

Polyps exposed to combined UVA and UVB were generally characterized by shorter, retracted, or absent tentacles and appeared lighter in color than control polyps (Fig. 3). Control polyps and polyps exposed to isolated UVA appeared qualitatively in similar health, with longer tentacles, evidence of feeding (presence of brine shrimp in the gut during and after feeding), a slightly darker pink/orange color (Fig. 3), presumably from consuming brine shrimp that also visually appear orange in color. Polyps exposed to isolated UVB still showed evidence of feeding, including presence of food in the gut, and were similar in color to control polyps and UVA only polyps, but often were observed with shorter or retracted tentacles (Fig. 3). Because it is difficult to accurately measure the size of scyphozoan polyps (see Lesniowski et al., 2015), quantitative size measurements were not performed. However, polyps exposed to combined UVA and UVB appeared visually smaller than polyps in any of the other treatments (Fig. 3) and had difficulty remaining attached to the experimental dishes.

Figure 3 Polyp appearance under UVR exposure.

Representative images of polyps under each UV treatment; (A) Control, Day 27; (B) UVA only, Day 27; (C) UVB only, Day 19; (D) UVA + UVB, Day 19. Treatment details are given in Table 1. Photographs by Lauren E. Johnson.

The proportion of dishes where polyps detached more than once during the study showed similar patterns compared to budding rate (Fig. 4). A chi-square test of independence confirmed that there was a significant difference in detachment across the UVR treatment groups (X2 (3, N = 54) = 22.51, p < 0.0001). A low percentage of the UVA only polyps detached from their dishes (8.3%), whereas 58.3% of control, nearly 66.7% of UVB only, and 100% of UVA + UVB polyps detached from their dishes. Post hoc comparisons revealed that significantly more UVA + UVB polyps detached from their dishes than any other treatment. In comparison, the percentage of polyps detached in both the UVA only treatment and the UVB only treatment did not statistically differ from the percentage of polyps detached in the control treatment.

Figure 4 Polyp detachment under UVR exposure.

The proportion (%) of dishes in each treatment where a polyp detached more than once during the 27-day study. Treatment details are given in Table 1.

Discussion

Here we present findings that illustrate the inability of Aurelia sp. polyps to survive and reproduce under UVR exposure that contains both UVA and UVB. Planulae demonstrate a strong preference for settling and metamorphosing into polyps on the shaded undersides of substrates (Svane & Dolmer, 1995; Pitt, 2000; Holst & Jarms, 2007). Multiple hypotheses have been proposed to explain this behavior (Brewer, 1976; Svane & Dolmer, 1995; Holst & Jarms, 2007), one of which proposes that planulae are avoiding solar irradiation (Svane & Dolmer, 1995). This study suggests support for that hypothesis by highlighting the potential consequences of UVR exposure on benthic polyps. Aurelia sp. polyps exposed to UVR containing both UVA and UVB did not reproduce and experienced 100% mortality within three weeks. In contrast, polyps that experienced no UVR survived and successfully reproduced.

Even though UVR reaching Earth’s surface contains both UVA and UVB, many studies assessing the consequences of UVR only consider UVB radiation because it is more energetic and harmful than UVA (Llabrés et al., 2013; Fischer & Phillips, 2014; Klein, Pitt & Carroll, 2016). Most marine organisms show sharp increases in mortality with increased exposure to UVB (Llabrés et al., 2013). Yet scyphozoan jellyfish polyps have been demonstrated to be relatively resilient to UVB alone, surviving and reproducing, but not necessarily at optimal rates (Klein, Pitt & Carroll, 2016), and our study is consistent with these findings. Polyps exposed to isolated UVB radiation survived and reproduced, albeit slower than either polyps exposed only to UVA or those not exposed to UVR at all (Fig. 2). However, UVB polyps were characterized as having retracted tentacles and reduced feeding, indicating reduction in overall health (Fig. 3), and over 60% of polyps exposed to isolated UVB detached from the substrate, which would result in mortality in a natural population.

We also quantified polyp response to isolated UVA, which can penetrate nearly two to three times deeper into the water column than UVB (Tedetti & Sempéré, 2006) yet is often overlooked in studies testing UVR stress (e.g., Fischer & Phillips, 2014; Klein, Pitt & Carroll, 2016). In the current study, polyps exposed to isolated UVA had the highest reproduction rate and lowest detachment from the substrate (Figs. 2 and 4). They were also characterized as having good health, demonstrated by extended tentacles and active feeding, similar to observations in the control treatment that was shielded from UVR. Medusae may be similarly resilient to isolated UVA; freshwater medusae exposed to UVA experienced no mortality in the short term (80 min; Salonen et al., 2012). Unfortunately, the UVA intensity we could achieve experimentally was lower than local ambient UVA levels (Table 1). As such, we caution extrapolating these results to how polyps would respond to UVA in shallow coastal waters where UVA intensities would be higher than demonstrated in this study. Although isolated UVA exposure may have seemingly minor effects, this area is ripe for future research, including the effects of naturally occurring levels of UVA (as opposed to the lower than ambient levels of UVA achieved in this study; see Table 1) and consequences of long term (days to weeks) exposure.

Despite achieving lower than ambient levels of UVA in this study, when exposed to a combination of UVA and UVB, Aurelia sp. polyps were unable to survive or reproduce. By day 21, all polyps exposed to combined UVA and UVB were deceased, and every polyp experienced multiple detachments from the substrate throughout the experiment. These polyps were also characterized as having retracted tentacles and reduced feeding as observed with UVB polyps, but ultimately ceased feeding and had absent tentacles. Polyps under combined UVA and UVB reduced into a small inactive white cyst before eventually dying (Fig. 3). Although polyps in this study survived for multiple days under combined UVA and UVB, in experiments on freshwater medusae, all medusae perished within just one hour of exposure to natural solar radiation containing both UVA and UVB (Salonen et al., 2012). If polyps had been exposed to higher (ambient) UVA levels, we would expect these results to be even more severe, potentially leading to more rapid mortality.

Many organisms have adapted mechanisms to cope with UVR exposure, including photoprotective compounds such as MAAs (Banaszak & Trench, 1995; Dunlap & Shick, 1998; Torres-Pérez & Armstrong, 2012), and pigments like melanin or carotenoids to mitigate UV effects (Rastogi et al., 2010). MAAs have been identified in a variety of marine organisms and act as a “sunscreen” for UVR (Nakamura, Kobayashi & Hirata, 1982; Dunlap & Chalker, 1986). MAAs originate in marine producers but can be found in a suite of invertebrates through relationships with MAA containing symbionts (Banaszak & Trench, 1995; Dunlap & Shick, 1998; Torres-Pérez & Armstrong, 2012). While symbiotic cnidarians often acquire MAAs through their symbiont, non-symbiotic cnidarians, as observed in the anemone Anthopleura elegantissima, likely receive MAAs from diet (Banaszak & Trench, 1995). As Aurelia sp. polyps do not contain symbionts and fed exclusively on brine shrimp in this study, we would not expect these polyps to have the protection from MAAs observed in other scyphozoan polyps, such as the upside-down jellyfish Cassiopea sp. (Klein, Pitt & Carroll, 2016). Other pigments such as melanin provide protection from both UVA and UVB (Kollias et al., 1991) and have been observed in scyphozoans, but melanin is often difficult to characterize due to a lack of defined structure or spectrum (Berking et al., 2005). Scyphozoans are capable of melanin production through the degradation of the amino acid tyrosine in the process of strobilation (Rastogi et al., 2010). The presence of melanin may assist ephyrae in the water column and alleviate potential UV-related effects, however, melanin production is restricted to the ephyrae and lacking in polyps (Berking et al., 2005), which would leave polyps vulnerable to UVR effects as the results of this study demonstrate.

Another strategy for dealing with UVR is avoidance, which, in the water column could be in the form of vertical migration. This strategy has been observed in a freshwater hydromedusae undergoing large diel vertical migrations to potentially limit exposure to harmful effects of UVR near the surface (Salonen et al., 2012). Benthic polyps, however, are restricted to settle on hard substrates. To overcome this, polyps are often observed on substrate habitats such as the sides or bottom of overhanging ledges (Lucas, Graham & Widmer, 2012) that are potentially shaded from direct UVR exposure. Although polyps could avoid UVR by settling on shaded substrates, Gleason & Wellington (1995) demonstrated that planula larvae of corals found at lower natural levels of irradiance were more sensitive to damaging effects of UVB. This strategy could, therefore, eliminate the evolutionary pressure to adapt UVR mitigating strategies and ultimately make polyps more sensitive to exposure to direct UVR as performed in this study. Furthermore, although polyps are able to move short distances (a few mm.) once attached to a hard substrate (pers. obs., L. Treible, 2019), they are unlikely to be able to migrate large enough distances to escape UVR exposure if planulae settle in a non-shaded habitat.

Planuae show high preference for the undersides of substrates (Pitt, 2000; Holst & Jarms, 2007), potentially to avoid solar irradiation (Svane & Dolmer, 1995). Yet, polyps are observed on many artificial substrates (Duarte et al., 2013; Makabe et al., 2014; Feng et al., 2017) in relatively shallow coastal environments, and the vertical surfaces of these structures likely still experience UVR exposure. Understanding the long-term effects of UVA on polyp health could help explain the behavior and success of polyps at depths where only UVA can penetrate, especially as ocean sprawl and the increase of artificial substrate along coastlines will increase potential polyp habitat (Duarte et al., 2013; Makabe et al., 2014; Feng et al., 2017).

Conclusions

In this study, we compared survival and reproductive success of Aurelia sp. polyps exposed to UVA, UVB, and both UVA and UVB in combination. The complete mortality of Aurelia sp. polyps exposed to both UVA and UVB radiation illustrates the importance of considering both UVA and UVB together when assessing organismal response to UVR stress, as only examining UVB underestimates the full impact of UVR that includes UVA. The inability of polyps to survive and reproduce under exposure to UVR could, in part, explain why the planula larval stage preferentially settle on the undersides of substrates where the resulting polyps are shielded from solar irradiation. Future studies should investigate the mechanisms that allow some polyps to survive and reproduce on vertical substrates exposed to solar irradiation without suffering the negative reproductive and survival consequences of exposure to UVR.

Supplemental Information

Supplemental Information 1 Results of two-way ANOVA

Comparison of the effects of UVA and UVB on Aurelia aurita polyp weekly budding rate (buds week−1). Values in bold are significant at p < 0.05. df = degrees of freedom.

Click here for additional data file.

Supplemental Information 2 Results of Tukey HSD pairwise comparisons

Comparison of Aurelia aurita polyp weekly budding rate (buds week−1) across UVR treatments. AA = UVA only; BB = UVB only; AB = UVA + UVB. Values in bold are significant at p < 0.05.

Click here for additional data file.

RAW DATA AND CODE

https://doi.org/10.5281/zenodo.7416874

We sincerely thank D. Gleason for lab space, equipment, and mentoring, J. Colón-Gaud for additional research support, and the John G. Shedd Aquarium (Shedd Aquarium Society) in Chicago, IL for cultures of Aurelia sp. polyps.

Additional Information and Declarations

Competing Interests

Author Contributions

Data Availability

The authors declare there are no competing interests.

Lauren E. Johnson conceived and designed the experiments, performed the experiments, analyzed the data, prepared figures and/or tables, authored or reviewed drafts of the article, and approved the final draft.

Laura M. Treible conceived and designed the experiments, performed the experiments, prepared figures and/or tables, authored or reviewed drafts of the article, and approved the final draft.

The following information was supplied regarding data availability:

Raw data and code available at Zenodo:

Johnson, LE. (2022). ljohnso14/JellyfishUVR: v.1.0.1 (v1.0.1). Zenodo. https://doi.org/10.5281/zenodo.7416874.

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
