# Peer review of "Hanging under the ledge: synergistic consequences of UVA and UVB radiation on scyphozoan polyp reproduction and health"

_PeerJ, doi:10.7717/peerj.14749_

## Round 0.1 · original submission · Major Revisions

The two expert reviewers and I think that the topic is of interest and your work has merit. However, concerns have been raised, which would preclude publication of the paper in its present form. In particular, the study of UVR mitigation mechanisms in general, and the choice of using carotenoids and MAAs specifically could be more comprehensively contextualized in the introduction. Further, the manuscript might benefit from more detailed reporting of data on polyp health and detachment, as well as their more detailed integration into the general discussion.

Both reviewers have provided very constructive feedback which we hope will help shape a revision of this manuscript. Please note that reviewer #1 has provided an annotated PDF document with more specific comments.

Reviewer 1 ·

Basic reporting

The study is an interesting work on the effects of UVR on polyps of the jellyfish Aurelia sp. which is a cosmopolitan scyphozoan jellyfish. I found the focus on how independent and combined UVA and UVB affect the survival and asexual reproduction of polyps to be a very interesting part of this paper. This finding was well supported by a correct experimental design with an appropriate number of replicates.

Other aspects of the paper were less well-integrated or contextualized in the paper. The introduction did well to mention the knowledge gaps in the study of the effects of natural solar radiation. However, the study of the mechanisms to mitigate UVR damage is often short in contextualizing the reasons why MAAs and carotenoids have been chosen to be measured in a non-symbiotic jellyfish species or genus.

Also, the number of replicates used for MAAs analyses and the wild jellyfish species used to compare the presence of these compounds need to be further justified in the methodology section and further supported by the literature. Similar improvements are needed to contextualize the analysis of carotenoids. Finally, the experimental design made for polyp migration study remains too short to obtain a consistent result.

Experimental design

Methods are well described with sufficient detail and information to be replicated. Contextualization of the research questions needs further work, especially in the experiments analyzing MAAs and carotenoid content, as well as the experiment on polyp migration. The mean research question (effects of UVR on polyps) is relevant and fills an identified knowledge gap. However, the already mentioned research questions (MAAs, carotenoids and migration) need further work.

Validity of the findings

All the underlying data have been provided. The impact of the findings and the conclusions of the work should be further described.

Additional comments

Please, find attached in the PDF file the general and specific comments that may contribute to improving the paper.

Annotated reviews are not available for download in order to protect the identity of reviewers who chose to remain anonymous.

·

Basic reporting

This paper reads well and is easy to understand. The authors provide novel insight into the combined effects of UVA and UVB radiation on the survival and proliferation of Aurelia sp. polyps, including pigment analyses of both polyps and ephyra for potential mechanisms to mitigate negative effects.

Experimental design

Overall the experimental design is described well. Hypotheses are clearly defined and followed up on throughout the paper. The authors decided to use UV radiation levels measured inland, near Statesboro, Georgia, to carry out their experiment, and already explain the underrepresentation of UVA in their study based of on these measurements. However, I suggest the authors include more detail on what ‘natural’ levels of UVA and UVB radiation might look like to Aurelia polyps, giving the general reader an indication to what extent this study might be representative of natural conditions polyps experience.

Validity of the findings

The authors manage to contribute new clear evidence of the combined detrimental effect of UVA/UVB radiation on Aurelia polyps, emphasizing the need for including UVA when studying the impact of UV radiation on marine organisms. While the discussion is short and concise, I challenge the authors to explore in more detail why UVA and UVB in combination might have this detrimental effect when UVA alone (in this study, at least) appear to have a neutral or positive impact on polyp health.

Additional comments

The following sentences require additional information in the text:
Line 154-155: We occasionally topped up the tanks with freshly made artificial seawater (35) to replace volume lost to evaporation.
Did the authors mean ‘topped up with freshwater’?

Line 220: After completing polyp observations, we fed the polyps freshly hatched brine shrimp…
The authors use presence of food item in polyps’ stomachs (active feeding) as a proxy for health, however when polyps are being assessed 24-48 hours after being fed this would lover the chance of polyps still containing food? Can the authors provide information if it is normal for polyps take more than 24 hours to digest prey?

Line 234: ..low energy UVB through UVA..
UVB have more energy (than UVA) and shorter wavelength, is this what the authors mean here?

Line 334-335: Neither the polyps nor the ephyrae from the laboratory culture of Aurelia sp. contained MAAs in their tissues…
The authors mention that animals from the laboratory culture were used to asses MAA content, while a wild caught S. meleagris was used to represent an adult medusae. We know light can stimulate growth of other light-harvesting or absorbing pigments in cnidarians, do the authors think MAA would have been present had the culture animals been pre-exposed or acclimated to natural doses of UV radiation?
It is not indicated in the methods whether culture animals were exposed to (UV) light.


Minor comments/suggestions:
Several times throughout the manuscript and in figure 2 species names (meleagris and aurita) have been written with capital.
While the authors provide intensities in table 1 for the UVA and UVB bulbs used for this study, it would be beneficial if a supplementary figure with the spectral information could be provided.
It was not clear to me at first that the UV light was provided by fluorescent tubes, perhaps mentioning this in the general setup of the experimental tanks, or provide a small figure with the setup would improve the clarity.
The Lexan polycarbonate sheet used to filter out UV light should include more detail on the UV-blocking properties, perhaps a manufacturer product sheet can be linked to?

---

## Round 0.2 · Minor Revisions

The reviewer is now satisfied with the authors’ point-to-point response. However, I noticed the authors are now introducing ‘synergistic’ consequences of UVA and UVB in the revised title. The authors have not referred to synergistic effects anywhere else in the manuscript before or after the revision. I apologize for not noticing this earlier, but going back to the description of the statistical procedures applied in this work, I have concluded it would be more appropriate to run a two-level ANOVA instead of the one-way ANOVA to assess the four levels of treatment (no UVR; UVA; UVB; combined UVA+UVB) on polyp budding. I also encourage the authors to provide the statistical tables as supplementary files, and to add more detail to the statistics section (were assumptions for ANOVA tested and met, etc). I am happy to consider the manuscript pending these final changes.

Reviewer 1 ·

Basic reporting

The revised manuscript is written clearly, and the literature and references are well-addressed. The results are very interesting and the results are shown in a professional way.

Experimental design

.

Validity of the findings

.

Additional comments

.

---

## Round 0.3 · accepted · Accept

Thank you for addressing the final comments. Merry Christmas!